# Secular Changes in Body Build and Body Composition in Czech Preschool Children in the Context of Latent Obesity

**DOI:** 10.3390/children8010018

**Published:** 2020-12-31

**Authors:** Petr Sedlak, Jana Pařízková, Daniela Samešová, Martin Musálek, Hana Dvořáková, Jan Novák

**Affiliations:** 1Department of Anthropology and Human Genetics, Faculty of Science, Charles University, Viničná 7, 120 00 Prague 2, Czech Republic; parizkova.jan@gmail.com (J.P.); daniela.samesova@natur.cuni.cz (D.S.); jan.novak@natur.cuni.cz (J.N.); 2Division of Child Health Promotion, Department of Hygiene, Third Faculty of Medicine, Charles University, Ruská 87, 10 00 Prague 10, Czech Republic; 3Department of Kinanthropology and Humanities, Faculty of Physical Education and Sport, Charles University, José Martího 269, 162 52 Prague 6, Czech Republic; musalek.martin@seznam.cz; 4Department of Physical Education, Faculty of Education, Charles University, Magdalény Rettigové 4, 116 39 Prague 1, Czech Republic; hanna.dvorak@seznam.cz

**Keywords:** body composition, body mass index (BMI), latent obesity, secular trend, preschool children

## Abstract

Changes in lifestyle can be significantly reflected in growth and development. Adaptations to reduced levels of physical activity, together with non-corresponding nutritional intakes, can result in body build and body composition changes at an early age. The present cross-sectional study aimed to evaluate the secular trend of modifications of body composition and body mass index (BMI) in Czech preschoolers over the last two to three decades. Boys and girls (386 boys and 372 girls) aged 4 to 6 years in 2014–2019 were measured. Outcome data were compared with the reference sample of preschoolers from 1990: 911 boys and 896 girls. Body height, BMI, and percentage of body fat, muscle, and bone mass were evaluated. Height and BMI have not changed. Body fat increased in both genders (*p* < 0.01), and contrarily, a significant reduction of muscle and skeletal mass was revealed (*p* < 0.001). Significant changes in body composition and unchanged BMI indicate the development of latent obesity during the last few decades. Due to latent obesity in a recent cohort, the differences in the prevalence of overweight and obesity markers according to BMI and fat percentage were tested. The prevalence of overweight and obesity was higher by 7.2% in boys, and by 6.5% in girls, as compared to children evaluated according to only their BMI results. Secular changes in preschoolers’ physical builds over the last 25 years are not reflected in body height and BMI, but in body composition. Insufficient development of active, lean body mass proportionally compensated by increased fat mass was also indicated.

## 1. Introduction

Changing environments and lifestyles have significantly impacted the growth and development of children and adolescents, furthermore reflecting a family’s health habits and standards from many points of view, starting from the very beginning of life [1,2]. An adaptation to reduced physical activity is one of the most decisive characteristics of modern lifestyles [3], resulting in several undesirable changes, most of which are permanent [4,5] and accompanied by increased health risks [6,7,8]. Systematic research of Czech preschool children began in the late fifties of the last century and has continued until the present; with the results summarized in the second edition of a special monography in 2010 [5]. Over the previous fifty years, these changes have been of concern to those studying all age groups, especially children and adolescents, starting at early preschool age [9]. Increasing adiposity and the deterioration of selected motor and other functional abilities have reduced the levels of physical activity in children and adolescents even further [8,9,10]. Furthermore, enhanced fat deposition on the trunk (evaluated by a skinfold thickness measurement) has also indicated an increased risk of metabolic syndrome, reduced cardiovascular fitness, musculoskeletal problems, etc. [11,12,13,14,15,16]. Inadequate nutrition, (i.e., energy intake not corresponding to real needs due to reduced energy expenditure) along with inappropriate dietary composition (i.e., an increased intake of simple sugars, saturated fats, etc.) have both been thought to contribute to overall health problems [17,18,19].

However, in preschool and prepubertal children, body mass index (BMI) has not shown any considerable secular changes, indicating that BMI does not completely reveal the above-mentioned response of body composition within changing environments [20,21,22,23]. Moreover, increased skinfold thickness and unchanged or even reduced circumference of the lower extremity indicates that in addition to fat, other components such as the possible unsatisfactory development of muscles and skeleton have also been occurring [9,24], and are accompanied by the deterioration of body posture, mode of gait, and the presence of pain in children. This was revealed by measurements of skinfold thicknesses, which did not alter or reduce the value of circumferential parameters when BMI remained in the span of normal body weight proportions [22].

Previous studies have focused their attention on specific body composition changes in the existing child population compared to those conducted several decades ago. They have been especially concerned by the increase in body fat and the apparent proportional decrease of active, lean body mass [9,24]. This shift is likely a reflection of environmental and lifestyle changes, especially a marked and decisive reduction of physical activity [3,4,11,12,13]. Early age corresponds to the highest levels of spontaneous physical activity, therefore limited physical activity from this age can have more significant immediate impacts as well as delayed undesirable effects. This is mostly due to an increased sensitivity on growth during this critical period of life, with adaptation to low physical activity developing later. Mostly inactive parents set poor examples for children, and therefore changes in family lifestyle [25], along with less security in public places available for children’s games, such as parks, nature, etc., all result in negative changes regarding child development, obesity, and deteriorated motor development, along with further decreases in spontaneous physical activity. Muscle mass, without proper stimulation through the adequate and dynamic exercise of the extremities—especially the lower ones, may be gradually replaced in part by increasing fat tissue, along with further functional and metabolic changes. This was revealed by measurements of skinfold thicknesses, which did not alter or reduce the value of circumferential parameters when BMI remained in the span of normal body weight proportions [22]. This state was defined as latent—“hidden” obesity, which has not been satisfactorily followed up during early childhood in available studies. Furthermore, the specific studies concerning the body composition and body development of normal, healthy, preschool children have been rare and mostly focus on BMI, and this also applies to older age categories [26].

Therefore, as a further line of future research, additional measurements of growth and evaluations of body composition particularly concerning fat, skeletal, and muscle development have been conducted since the beginning of the nineties in the last century up to the present day. As indicated by the above-mentioned studies, it is indispensable to evaluate additional somatic adaptational variations as significant markers of not only present, but also possible future health complications, such as, obesity, dyslipoproteinemia, increased blood pressure, diabetes, etc.

The aim of this study was to describe and evaluate the changes of the secular trend of body build (BMI, body height) and body composition (percentage of body fat, muscles, and skeleton mass) in healthy Czech preschool children, between recently collected data (2014–2019), and reference samples from 1990. This can enable a more exact evaluation and prognosis of children’s health risks for the future, and from an early age.

## 2. Materials and Methods

### 2.1. Participants

This cross-sectional study of preschool children was conducted in continuity with the pilot study where were observed significant changes in the thickness of skinfolds together with paradox non-significant changes in BMI [24]. Children of normal, middle-class families corresponding to comparably similar European families were included. From 2014 to 2019, anthropometric measurements were obtained from 758 children (386 boys and 372 girls) who were from ages four to six and living in the capital city of Prague with comparable lives and environmental conditions. Resulting data were compared with the Czech reference sample from 1990 provided by Bláha [27]. Children were measured in all parts of the Czech Republic. Data ascertained in this sample have remained valid for the Czech preschool population as criteria of reference.

The present research was approved by the Ethics Committees of the Faculty of Science, Charles University, Prague (approval number 2017/23). The study was conducted in accordance with the Declaration of Helsinki (Fortaleza actualization). Written informed consent was obtained from parents of all children participating in the study. Parents agreed to the anthropometric measuring of their children, in cohort 2017–2019, and a bioelectrical impedance analysis examination. All data was anonymized.

### 2.2. Measurements

Children were examined at their kindergartens, always during the morning hours, and wearing only their underwear. Health status was regularly checked, and children with even minor problems were excluded (such as elevated body temperature, coughs, colds, etc.). Suitable corresponding parameters were ascertained by the same method in the reference group followed up in 1990. Anthropometric measurements were conducted according to standard anthropometric techniques [28]. Body height was measured using a portable stadiometer Trystom (Trystom, s.r.o., Olomouc, Czech Republic; exactitude 1 mm). Body mass was ascertained during bioimpedance analysis (BIA) assessments, while in the reference group followed up in 1990, it was ascertained by a stamped weight (precise to one-tenth kg). Body mass index (BMI) was calculated as BMI = body mass (in kilograms)/ body height^2^ (in meters).

The Matiegka modified method of anthropometric assessment [29,30] was utilized in the 1990 cohort for the analysis of body composition. To enable comparisons between the two cohorts, the same assessment method was used for the 2014–2019 cohort. Participants in the recent sample were also measured with bioimpedance analysis (BIA) assessments, which allowed for a comparison of both methods and the estimation of body composition components using Matiegka equations. Matiegka method was selected as it is a valid method for the juvenile population [30] and also allows for the analysis of not only body fat mass, but also muscle and bone masses. Matiegka method was based on the measurement of body height and weight, breadth and circumferential measurements, and skinfold thicknesses. The resulting regression equation gives the total body fat, muscle, and skeletal mass in both kilograms and body weight percentage. The agreement between Mateigka’s method and BIA in the estimation of fat mass and muscle mass was tested using the Bland–Altman method [31].

All anthropometric parameters were ascertained using standard methods on the right side of the body, breadth to the exactness of 1 mm [28]. Breadth measures characterizing skeletal robusticity were measured by contact rule—i.e., biepicondylar breadth of humerus, biepicondylar breadth of the femur, bistyloideal breadth, and bimalleolar breadth. Circumferential measurements were conducted with a tape measure. The circumference of the relaxed arm was measured at the half-way point between the acromion and the olecranon, and the medial circumference of the thigh was measured at the half-way point between the trochanter and the tibiale anthropometric points. The thickness of skinfolds was measured with a modified caliper type Best (constant compressive strength of the tips = 2 N), on the chest (above the 10th rib at the point of intersection with the anterior axillary line), in the abdominal region, on the upper arm (above the triceps and biceps brachii), then also on the forearm, on the mid-thigh, and on the calf (over the line of maximal circumference).

The calculation of body composition according to Matiegka equation is as follows [29,30]:

Ratio percentage of skeleton mass:(1)$$O=o^{2}*h*k_{1};\;o=(o_{1}+o_{2}+o_{3}+o_{4})/4$$

*o*_1_—biepicondylar breadth of humerus, *o*_2_—biepicondylar breadth of femur, *o*_3_—bistyloideal breadth, *o*_4_—bimalleolar breadth, *h*—body height, *k*_1_ = 1.2.

Ratio percentage of body fat:(2)$$D=d*S*k_{2};\;d=\frac{1}{2}*\frac{d_{1}+d_{2}+d_{3}+d_{4}+d_{5}+d_{6}}{6}$$

*d*_1_—skinfold above biceps, *d*_2_—on the forearm, *d*_3_—on the mid-thigh, *d*_4_—on the calf 2, *d*_5_—on the chest 2, *d*_6_—on the abdomen, *S*—body surface: 71.84 * body mass^0.425^ * body height^0.725^, *k*_2_ = 0.13.

Ratio percentage of muscle mass:(3)$$M=\;r^{2}*h*k_{3};\;r=\frac{r_{1}+r_{2}+r_{3}+r_{4}}{4}$$

*r*_1_ = (relaxed arm circumference/π) − (skinfold above biceps/2) − (skinfold above biceps/2), *r*_2_ = (maximum forearm circumference/π) − forearm skinfold, *r*_3_ = (medium thigh circumference/π) − skinfold on the mid-thigh, *r*_4_ = (maximum calf circumference/π) –skinfold on the calf, *h*—body height, *k*_3_ = 6.5.

Bioimpedance analysis (BIA) was conducted using InBody 230 (DMS-BIA technology; InBody Co., Seoul, Korea), which is declared by the manufacturer to be applicable for children from the age of 3 to estimate body composition (body fat and muscle percentages).

### 2.3. Statistical Analysis

The data were analyzed using Statistica software, version 9.0, and R programming language version 3.6.1 (R Core Team) using Tidyverse (version 1.3.0) and blandr (version 0.5.1) packages. Before performing robust statistical tests, the normality of each sample distribution was tested and non-normally distributed data were transformed towards normality with the Box-Cox method. The secular trend of height, BMI, and components of body composition (by Matiegka method) were tested using robust parametric tests. Differences between the results of the used methods of body composition estimation (Matiegka, BIA) were tested using a two-sample t-test and Bland–Altman analysis. Bonferroni correction for multiple comparisons was applied to prevent false significance. Along with statistical significance, the clinical validity of differences using Cohen’s d was verified (using a classification of effect size as 0.2—small, 0.5—medium, 0.8—large). Classification proprieties of body weight categories (under 10th percentile, between 10th and 85th percentile, and above 85th percentile) were estimated using BMI and body fat percentages were compared and tested with the Chi-square test for categorical variables. Statistical significance was assessed at the level of *p-*value ≤0.05, ≤0.01, and ≤0.001.

## 3. Results

Results concerning body height, BMI, and body composition components as a percentage of total body mass (body fat, active body mass) were compared with the reference database of Czech preschool children followed up in 1990.

With regard to body height, no significant differences due to the age categories of both genders were found (Table 1 and Table 2). This finding confirms the disappearance of the secular trend of height increase in the Czech population. The same conclusions concerned body proportionality evaluated by BMI. There were no significant differences in any age category in the girls, the BMI values showed comparable values for both cohorts (Table 2). Likewise, in the boys the differences in BMI were very low (Table 1). A significant difference was observed only in a cohort of boys aged five, which was a decrease in BMI values over time (*p* < 0.05).

Significant changes in body composition were found between the present and reference samples, which were assessed with Matiegka method. The percentage of body fat was significantly higher in both gender and age groups. These differences were clinically relevant (Cohen’s d = 0.4–0.5). Opposite results concerning active body mass were also observed (Table 1 and Table 2, Figure 1). The percentage of muscle and skeleton tissue was markedly decreased (*p* < 0.001), with a high clinical relevance (size effect by Cohen’s d = 0.4–1.0).

In the most recent sample, body composition was evaluated using BIA, and the agreement between the results of both methods was tested. Significant differences in all components of body composition among all age groups of boys were found (*p* < 0.05). In girls, significant differences in the percentage of body fat of 4-year-olds (*p* < 0.01) and in the percentage of muscle mass in 5 and 6-year-olds (*p* < 0.001) were found. In all cases, Matiegka method underestimated these results. The corresponding trend has already been observed in a group of children with obesity when comparing this method to the results of dual-energy X-ray absorptiometry [32]. Based on Bland–Altman analysis, overall agreement between BIA and Matiegka method was higher in the case of fat mass (Figure 2) estimation than muscle mass (Figure 3) estimation, with an average bias value −0.18 (CI 0.56 to 0.20; SD = 4.33), and 1.99 (CI 1.63 to 2.36; SD = 4.08), respectively.

Due to the occurrence of latent obesity, the differences in the prevalence of individual categories (low body mass < 10 percentile, normal body mass 10–85 percentile, excess body mass > 85 percentile), in BMI and percent of body fat evaluated by using the BIA method, were tested using the Chi-square test. In the category below the 10th percentile, the results were in agreement. In the central category, the results corresponded on approaching significance (*p* = 0.056 in girls, 0.011 in boys, respectively). Above 85th percentile, the differences were significant (*p* < 0.05) (Table 3). In the sample of recently measured children evaluated according to body fat, the prevalence of overweight and obesity was higher by 7.2% in boys, and by 6.5% in girls, as compared to children evaluated according to their BMI values only.

## 4. Discussion

The study aimed to evaluate secular modifications of somatic development, including body composition, in preschool-aged children. These modifications have yet to be analyzed and confirmed. The comparison of recent data and those ascertained in 1990 concerning body height in preschool children revealed the attenuation of the secular trend of growth acceleration. Studies that compared the growth of Czech children from birth until 18 years of age between the years 1951 to 2001 confirmed a slow-down of secular increase in body height, especially in prepubertal growth categories [33]. The stabilization of environmental factors such as nutrition, healthcare, etc., indicates that the maximal level of genetic growth potential has been, at present, generally achieved; therefore, the average height of the population no longer changes, which also includes preschool children. This was also revealed in other child populations [34,35,36,37,38].

However, this does not concern body composition—the proportion of individual components of the body. Body fat percentages, in particular, have increased, and the distribution of them has changed too [9,24,39]. Results of other long-term studies concerning body composition and its secular trend in other preschool populations are rarely included in longitudinal studies. Moreover, such a trend in Czech preschool children has not been reflected in a change of BMI since the 1990s. This finding confirms that changes in body composition cannot be detected using solely BMI evaluation. Regarding the secular change of BMI in other populations, almost no measurements or evaluations of simultaneous body composition change have been included [20,21,22,23].

These observations indicate the manifestation of so-called latent, or “hidden”, normal-weight obesity, which has also been revealed as present in the adult population, and can be accompanied by serious functional, metabolic, and cardiovascular problems [40,41]. The incidence of metabolic syndrome in adult normal-weight obese subjects was four-times higher than in the normal population [42]. Latent, normal-weight obesity may also be observable in young children, in school age and was also found in adolescents [43,44]. These results indicate an increased functional, metabolic, and overall health risk of an increased proportion of fat—evidently at the expense of other tissue development—mainly stimulated by the adaptation to reduced physical activity, uncharacterized by marked modifications of BMI [45], and can be satisfactorily detected by further somatic characteristics in this age group.

Therefore, special attention has to be focused on the evaluation of simultaneous changes of body composition (modifications of the size and character of muscle and skeletal tissues) which are the main causes of negative adaptation to hypokinesia characteristics of present lifestyle habits [3,4,46,47]. In this respect, it would be extremely important to also have data on secular changes of more detailed evaluations of body components, such as vital organs and their possible modifications and changes, as they are defined, for example, in the case of the impact of long-term reversals like increased physical activity through exercise, various sports, etc. These results have been found mostly for adults and adolescent athletes, with systematic studies concerning preschoolers not yet available.

Similar research has also not focused on opposing circumstances, i.e., the impact of long-term adaptations to the systematic reduction of physical activity—which impedes natural developmental characteristics at an early age. This concerns the changes in body composition—not only fat, but also muscle, skeletal, and especially vital organ modifications, during early growth. Adaptation to changes in long-term energy balance and turnover is a factor that is essential in this respect, as short-term hypokinesia does not result, especially in morphological consequences (skeletal and muscular [48,49]), but may manifest in metabolic, biochemical, and functional changes which may be transitional. A reduction in physical activity also results from general changes of lifestyle such as the reduced availability of open places, backgrounds, parks, and free nature, and the increased use of computers, mobiles, and TV all from a very early age [13,25]. Special attention has recently been paid to the development of further abilities, especially motor functions starting at an earlier age, and also as relates to actual and future health status [49,50,51,52,53,54]. The greatest health risks resulting from an adaptation to a long-term, stable reduction of physical activity during growth are functional, metabolic, and biochemical consequences, resulting ultimately in morphological consequences (i.e., body composition). As the proportion of total body fat increases along with altered distribution, muscle development is prevented from achieving its genetic potential—the same applies to bone development. This trend has been continuing to manifest itself even more markedly during the last few decades; however, studies including more complex views on this problem—using further functional, skeletal measurements—have not been available until present [9,24]. Secular changes related to low physical activity (which impacts vital organs) are also possible, especially those that affect the heart and respiratory system. The aforementioned consequences also influence the potential level of physical activity, creating a negative feedback loop, and further worsening the whole situation. This is risky, especially during growth—mainly during the early critical periods when the level of spontaneous physical development is at its highest level. The growing child is thus more sensitive to the impact of reduced physical activity, which particularly applies to long-term adaptations and all previously mentioned consequences. Energy imbalances and the low turnover of energy intake/output that occur at an early age are considered risks for later health problems. However, timely intervention may be one of the most desirable avenues of prevention and should start as early as possible at the beginning of one’s life.

The impact of insufficient physical activity, and especially the adaptation to it during long periods of life, is a characterization of present lifestyles which concerns the whole population and also includes school children and adolescents. The aforementioned results confirm that significant negative effects exist already in children of young ages and reveal in greater detail that undesirable morphological and functional consequences could manifest as health decline later in life, commencing already with preschoolers. The effects of reduced physical activity have been emphasized more recently, particularly during the last two to three decades. Without the proper intervention of pedagogues, public health policymakers, pediatricians, further medical specialists, etc., and parents—low physical activity starting in early childhood may lead to detrimental health consequences later in life.

Limitations of this study can be seen in the recruitment of children from the area of just the capital city of Prague. However, nation-wide anthropological surveys of children and adolescents (1991 and 2001) did not reveal any significant differences in body height and BMI between Prague and other Czech regions [55]. Other limitations are the methods used for the estimation of body composition. We used Matiegka’s original equations and also, in the case of the recent samples, bioimpedance analysis (BIA). Both approaches have their limitations. Anthropometric methods are dependent on the experience of the examinators, and results can also vary depending on the selected equation. BIA can be influenced by individual hydration. Other more precise methods also have significant limitations, for example, the use of dual X-ray absorptiometry or computer tomography can result in the exposure of proband to radiation, but even in the case of dual X-ray absorptiometry, this exposition is is still a complication for researchers even though it is minimal [56]. Magnetic resonance imaging, or plethysmography, are highly precise but not feasible in field studies.

It is essential to take into account that different methods for body composition estimation are not fully comparable and the monitoring of longitudinal changes or the comparison of diverse populations always have to use the same methods. From the perspective of the longitudinal monitoring of trends, it is fundamental to leverage precision and future replicability; in this context, anthropometry seems like a relatively useful tool.

## 5. Conclusions

Body height and BMI in Czech preschool children have not changed during last 25 years, but significant changes in body components have been found, along with an increase in body fat and a significant reduction of active, lean body mass (muscle and skeletal components). The situation was found to be more critical in boys, and the trend was stronger with age in both sexes. These changes in body components were found to extend in the same ratio, so the value was not reflected in the BMI. This finding indicates a detailed complex evaluation of body composition, revealing the development of latent obesity even in the case of an unaltered BMI, as this situation can lead to an increased risk of a decline in health status with age.

## Figures and Tables

**Figure 1 children-08-00018-f001:**
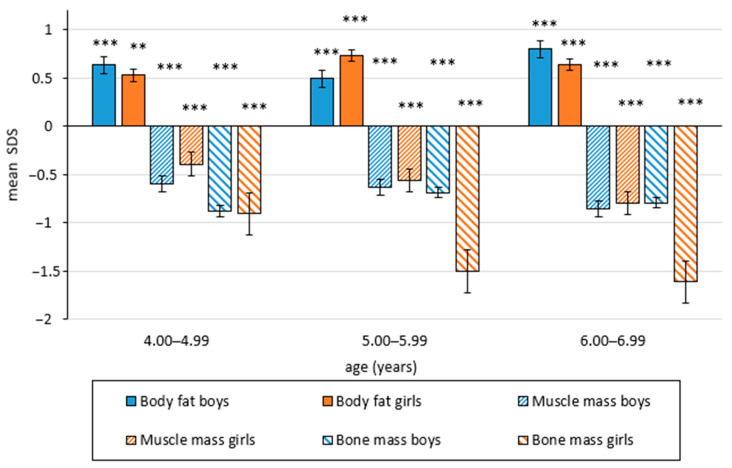
Difference between mean SD-score values of reference and present study sample for each body component (estimated using Matiegka method). Statistical significance level ** *p* < 0.01; *** *p* < 0.001.

**Figure 2 children-08-00018-f002:**
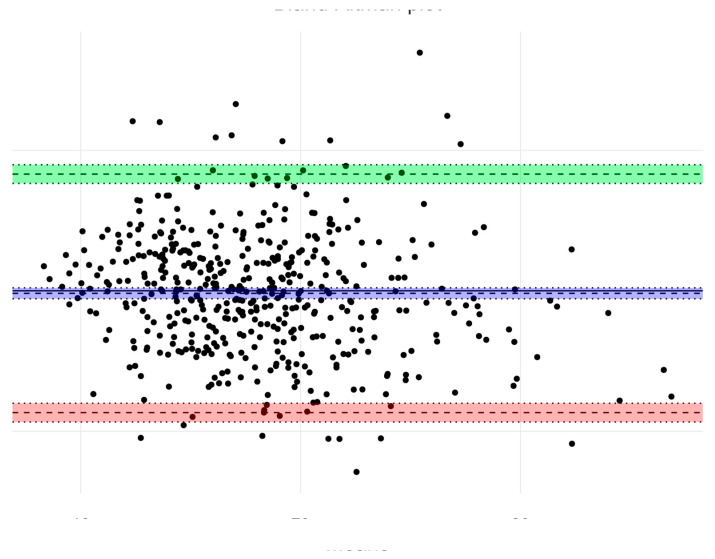
Bland–Altman Plot of agreement between fat mass estimation using bioimpedance analysis (BIA) and Matiegka method. The overall agreement is relatively high, with a mean bias value −0.18.

**Figure 3 children-08-00018-f003:**
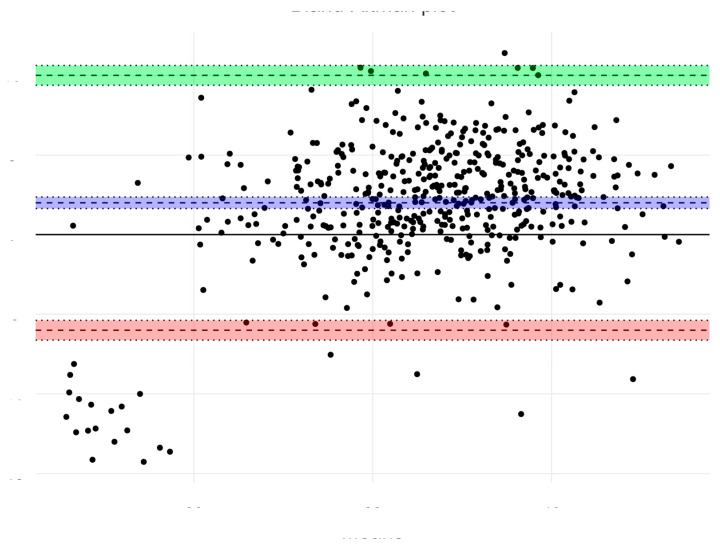
Bland–Altman Plot of agreement between muscle mass estimation using bioimpedance analysis (BIA) and Matiegka method. The overall agreement is average, with a mean bias value 1.99.

**Table 1 children-08-00018-t001:** Comparison of monitored somatic traits in present Czech preschool children aged 4–6 years with sample 1990—boys.

Age (Years)	4				5				6			
Sample	A (*n* = 115)	B (*n* = 322)			A (*n* = 140)	B (*n* = 369)			A (*n* = 131)	B (*n* = 220)		
	Mean (SD)	Mean (SD)	*p* Values	Cohen’s d	Mean (SD)	Mean (SD)	*p* Values	Cohen’s d	Mean (SD)	Mean (SD)	*p* Values	Cohen’s d
Body height (cm)	107.6 (5.13)	107.4 (4.79)	0.582	0.0	115.0 (5.07)	114.1 (5.01)	0.164	0.2	120.3 (5.24)	120.1 (4.86)	0.626	0.1
BMI (kgm^−2^)	15.3 (1.25)	15.5 (1.24)	0.896	0.0	15.2 (1.20)	15.5 (1.30)	0.023	0.2	15.3 (1.29)	15.6 (1.46)	0.095	0.2
Body fat (%) ^*)^	18.3 (4.02)	16.1 (4.18)	<0.001	0.5	17.5 (5.58)	15.6 (4.63)	<0.001	0.4	17.8 (6.50)	15.0 (5.10)	<0.001	0.5
Muscle mass (%) ^*)^	34.2 (3.19)	36.5 (3.72)	<0.001	0.6	35.6 (3.21)	37.5 (3.31)	<0.001	0.6	36.4 (3.16)	39.2 (3.52)	<0.001	0.8
Bone mass (%) ^*)^	19.1 (1.48)	20.3 (1.43)	<0.001	0.9	19.5 (2.00)	20.6 (1.65)	<0.001	0.7	19.6 (2.08)	21.0 (2.62)	<0.001	0.5

Legend: A = sample A: survey 2014–2019, *n* = 758 (372 girls, 386 boys); B = sample B: survey 1990, *n* = 1807 (896 girls, 911 boys); ^*)^ = estimated by Matiegka method.

**Table 2 children-08-00018-t002:** Comparison of monitored somatic traits in present Czech preschool children aged 4–6 years with sample 1990—girls.

Age (Years)	4				5				6			
Sample	A (*n* = 112)	B (*n* = 314)			A (*n* = 153)	B (*n* = 391)			A (*n* = 107)	B (*n* = 191)		
	Mean (SD)	Mean (SD)	*p* Values	Cohen’s d	Mean (SD)	Mean (SD)	*p* Values	Cohen’sd	Mean (SD)	Mean (SD)	*p* Values	Cohen’s d
Body height (cm)	106.6 (5.16)	106.8 (4.68)	0.776	0.0	114.2 (5.18)	113.6 (4.78)	0.744	0.1	119.1 (5.40)	119.5 (5.23)	0.665	0.1
BMI (kgm^−2^)	15.5 (1.26)	15.5 (1.37)	0.793	0.0	15.5 (1.56)	15.5 (1.63)	0.987	0.0	15.4 (1.52)	15.4 (1.58)	0.622	0.0
Body fat (%) ^*)^	20.9 (5.18)	19.0 (4.98)	0.002	0.4	22.2 (6.90)	17.9 (5.22)	<0.001	0.8	20.9 (6.09)	17.5 (5.49)	<0.001	0.6
Muscle mass (%) ^*)^	34.3 (2.92)	36.2 (3.97)	<0.001	0.7	35.3 (3.49)	37.4 (3.80)	<0.001	0.7	35.9 (2.99)	39.2 (4.16)	<0.001	0.8
Bone mass (%) ^*)^	17.6 (1.67)	19.1 (1.53)	<0.001	0.9	18.2 (1.55)	19.1 (1.72)	<0.001	0.5	18.1 (1.75)	19.8 (1.91)	<0.001	1.0

Legend: A = sample A: survey 2014–2019, *n* = 758 (372 girls, 386 boys); B = sample B: survey 1990, *n* = 1807 (896 girls, 911 boys); ^*)^ = estimated by Matiegka method.

**Table 3 children-08-00018-t003:** The differences in the prevalence of low, normal, and overweight by Body Mass Index (BMI) and percent body fat (χ^2^ test).

Cut off Value	Boys (*n* = 337), Age 4–6 Years			Girls (*n* = 338), Age 4–6 Years		
	BMI	% Fat	*p* Values	BMI	% Fat	*p* Values
<10 percentile	9.6 (6.5–12.8)	10.9 (7.6–14.2)	0.611	7.9 (5.0–10.8)	7.9 (5.0–10.8)	1.000
10–85 percentile	80.7 (76.5–84.9)	72.2 (67.4–77.0)	0.011 *	80.0 (75.5–84.1)	73.5 (68.6–78.0)	0.056
>85 percentile	9.7 (6.5–12.8)	16.9 (12.9–20.9)	0.009 **	12.1 (8.7–15.6)	18.6 (14.5–22.8)	0.025 *

In % (95% CI). Statistical significance level * *p* < 0.05; ** *p* < 0.01.

## Data Availability

Data are available in a publicly accessible repository.

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
