# Peer review of "Secular Changes in Body Build and Body Composition in Czech Preschool Children in the Context of Latent Obesity"

_children, 2020, doi:10.3390/children8010018_

Round 1

Reviewer 1 Report

There are some typographical and grammatical errors and in places appropriate academic language has not been used. I have attempted in the attached file to correct some of these incidences but have not highlighted all of them. Please proof-read the manuscript to further correct these. 

Reviewer 2 Report

Abstract

Line 19: “…over the last two to three decades; (386 boys and 372 girls) aged 4 to 6 years in 2014-19 were measured….”.  2014 to 2019 are two to three decades?

Introduction:

Line 30: What does mean “potential”?

Lines46-49. Articles cited to support this sentence study short periods, authors are talking about secular changes.

Lines 49-50: What sorts of studies? There are a lot of studies that use DEXA or advanced-BIA techniques.

Lines 51-53 Both papers cited by authors to support this asseveration [9 and 24] analyze the same sample (Czech children). Are the authors wide spreading this claim to every children population?

Lines 55-57. Again the authors cite the same papers (9 and 24).

Lines 57-59. The paper [3] probably is very general and unspecific to explain the behavior in children. There are a lot of investigations about this topic (body composition and physical activity in children).

Lines 59-62: “As an essential factor it is indispensable to consider and emphasize the start in early age when the level of spontaneous physical activity is mostly highest, and therefore limited physical activity can have a more significant presence as well as delayed undesirable effects.” Is an opinion of the authors?

Lines 62-67 “Mostly inactive parents set poor examples for children, and thus changes in family lifestyle, less security in public places available for children's games, parks, nature etc.,  result in negative changes in regards to child development, obesity, and deteriorated motor development,  along with further decreases in spontaneous physical activity”.

Is an opinion of the authors?

In my opinion, the authors use excessively two papers (9 and 24) to justify this article. And that is my main concern because these papers (9 and 24) seem very similar to this article, specifically:

Sedlak, P.; Pařízková, J.; Procházková, L.; Cvrčková, L.; Dvořáková, H. Secular changes of adiposity in Czech children aged from 3 to 6 years: latent obesity in preschool age. BioMed. Res. Inter. 2017, doi.10.1155/2017/2478461.

Lines 78-82: “Therefore, additional measurements of growth and evaluations of body composition concerning  especially fat, skeletal, and muscle development have been conducted since the beginning of the nineties in 79 the last century up to the present day. As indicated by the above-mentioned studies, it is indispensable to 80 also evaluate additional somatic adaptational variations as significant markers of not only present, but also 81 possible future health complications (obesity, dislipoproteinemia, increased blood pressure, diabetes etc.)”.

Is an opinion of the authors?

 The articles referenced to support the core of the investigation problem are very few, I recommend a better review.

  1. Materials and Methods

Line 89: If paper [24] is a pilot study, why the sample in the pilot study is bigger the main study?

Line 92: Could you give more details on the sample size calculation? Did you calculate significance level or power?

Line 93: Is the sample representative for Czech? this sample has been compared with a sample which represents hole Czech Republic, but in this study, the sample is just from Prague.

Lines 114-118: the wording and syntaxis are unclear

The main concern is that this paper is very similar to the next article:

Sedlak, P.; Pařízková, J.; Procházková, L.; Cvrčková, L.; Dvořáková, H. Secular changes of adiposity in Czech children aged from 3 to 6 years: latent obesity in preschool age. BioMed. Res. Inter. 2017, doi.10.1155/2017/2478461.

Similar objectives, similar sample (maybe same sample), similar results. It seems that is the same investigation with very minor changes, but the outcomes are the same or at least very similar. The authors must explain with details if is a different study and if this paper provides a significant and new contribution to the scientific area

Reviewer 3 Report

Although the authors have done a great job, I consider that the following points could be improved:

  1. Correspondence, double r
  2. Abstract, please developed results
  3. “participants” better than “sample”
  4. th in superindex
  5. Thought the text “body mass” better than body weight
  6. Inclusion and exclusion criteria?
  7. Include Helsinki Declaration en Fortaleza actualization
  8. Include reference in all devices
  9. Isaak technician include and number
  10. Ethical consideration moves on participants last part of the paragraph
  11. The p is italicized and lowercase, and there must be spaces.
  12. Is Bonferroni adequate in this type of analysis, t-test or chi square with effect size?
  13. Describe all results based on table n1 and n2
  14. In results part, no references
  15. Table 3, weight categories. Change title please
  16. You can include one graph comparing both genders
  17. Table 3, describes very interesting data about percentiles, discuss the main significant changes thought the discussion part
  18. Star discussion part with a general approach describing main goal and main obtained results
  19. Line 260-262, this sentence needs a reference
  20. Line 264, “this trend,,, include one reference to validate this idea
  21. Line 724, “ permanent situation of this sort”, I don’t understand this sentence, rewrite please
  22. Line 280. Delete conclusions, may be confuse, given that you have the last paragraph with conclusion idea
  23. Include future lines
  24. Include practical applications
  25. Include another different paragraph with limitations
  26. Line 301 spaces between bmi    in
  27. Line 310-317 letter size, uniform please
  28. Conclusion part shorter
  29. Second part, the impact off… is no conclusion
  30. reference number 5, 2nd in super index
  31. 27 idem 3 rd
  32. doi in all references
  33. Consider update references

 Dalrymple KV, Flynn AC, Seed PT, Briley AL, O'Keeffe M, Godfrey KM, Poston . Associations between dietary patterns, eating behaviours, and body composition and adiposity in 3-year-old children of mothers with obesity. L.Pediatr Obes. 2020 May;15(5):e12608. doi: 10.1111/ijpo.12608. Epub 2019 Dec 27.

  1. Consider update references

Kreissl A, Jorda A, Truschner K, Skacel G, Greber-Platzer S. Clinically relevant body composition methods for obese pediatric patients.

BMC Pediatr. 2019 Mar 21;19(1):84. doi: 10.1186/s12887-019-1454-2.

Reviewer 4 Report

The proposed manuscript by Sedlak et al. presents the results from a cross-sectional study aimed to investigate the BMI and body composition modification in Czech preschool children during two-three decades. Study involved a total of 386 boys and 372 girls, aged 4-6 y.o. As a reference, the authors used a sample of 1,809 preschool children (911 boys and 896 girls) from 1990. Data showed significant reduction of lean body mass and increase in body fat mass during the study time period as the changes in body components extend in the same ratio and the value was not reflected in the BMI. Thus, based on the obtained findings, the authors of the study highlighted the need of more complex evaluation of body composition in order to reveal the development of latent obesity.

I do not see any problems with the study design or the statistics. The study has some limitations, but they are accurately discussed by the authors.

Round 2

Reviewer 1 Report

The authors have made the required changes and the manuscript is significantly improved. I would like to thank the authors for taking the time to improve the manuscript. 

I would advise the authors to carefully proof-read the manuscript again as there a a few typos, grammatical errors and some repetition in places.

Author Response

Thank you for the re-review of our manuscript! We tried to clean the manuscript a bit, hope it is better now.

Best Wishes,

Authors

Reviewer 3 Report

Authors have made a great effort to improve the manuscript.

Author Response

(The authors gave the same response as above.)
